# Fabrication and Characterization of Ni60A Alloy Coating on Copper Pipe by Plasma Cladding with Induction Heating

**Jinjin Lv, Chao Zhang, Zhiyu Chen, Dan Bai, Yuwen Zhang \*, Guangshi Li and Xionggang Lu \*** 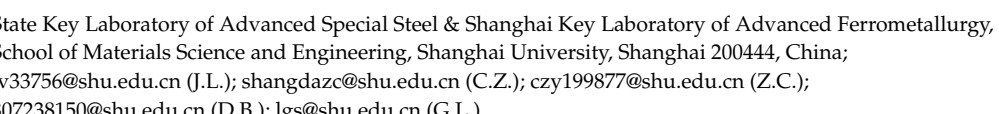

State Key Laboratory of Advanced Special Steel & Shanghai Key Laboratory of Advanced Ferrometallurgy, School of Materials Science and Engineering, Shanghai University, Shanghai 200444, China; lv33756@shu.edu.cn (J.L.); shangdazc@shu.edu.cn (C.Z.); czy199877@shu.edu.cn (Z.C.); 807238150@shu.edu.cn (D.B.); lgs@shu.edu.cn (G.L.)

\* Correspondence: springzyw@shu.edu.cn (Y.Z.); luxg@shu.edu.cn (X.L.)

**Abstract:** Plasma cladding coupled induction heating was developed and successfully used to fabricate Ni60A coating on the surface of copper pipe. By matching the swing arc with the rotating copper pipe, the cladding efficiency was as high as 32.72 mm$^2$/s. From the head to the tail of the coating, the wear resistance changed from 4.5 to 1.8 times that of pure copper. During the cladding process with constant current, the surface temperature of the cladding zone and the bath depth gradually increased. The corresponding dilution ratio increased, accompanied by the widening of the interface transition zone and the growth of precipitated phases (CrB and Cr$_{23}$C$_6$). Due to the gradient change of composition, the coating can be regarded as an in situ synthesized gradient coating. The critical point of sudden change of temperature in cladding zone was 850 °C, at which point the wear mechanism changed from abrasive wear to adhesive wear. The proper surface temperature of cladding zone should be controlled within 600–850 °C, which can be achieved by matching the cladding current and induction heating power. Results indicated that plasma cladding coupled induction heating is a potentially effective method to prepare high-quality coating on the surface of a large-complex-curved copper component.

**Keywords:** copper pipe; surface strengthening; Ni60A alloy; plasma cladding; induction heating





## 1. Introduction

Steel is the most widely used metal material in the world [1–3]. At present, more than 75% of steel is produced from iron ore by the blast furnace-converter long process. The main process equipment for reducing iron ore to molten iron is the blast furnace. As the largest steel producer in the world, China has more than 1000 blast furnaces [4]. The water-cooled tuyere small sleeve is the core component of the blast furnace, and its function is to send hot air and inject pulverized coal. The material used for making tuyere small sleeve is pure copper, which has high thermal conductivity. However, poor wear resistance is the main drawback for copper [5–8]. The front end and inner wall of the pure copper sleeve are easily worn by high-speed moving coal and coke, resulting in leakage and reduced service life [9]. Frequent damage and replacement of the sleeves has extremely adverse effects on the blast furnace. Therefore, it is necessary to fabricate a wear-resistant coating on the surface of the copper sleeves to prolong their service life.

In recent years, the surface coating technologies including electroplating, cold spraying, thermal spraying, laser cladding and plasma cladding are used to improve the wear resistance of copper [10–14]. Among the surface engineering technologies, plasma cladding has the unique advantages of low cost and high efficiency, which results in a different coating with metallurgical bonding and high performance on a copper substrate. Li et al. [15] prepared the Ni-based (Ni25) wear-resistant layer by plasma cladding after preheating the copper plate. The average microhardness of the wear-resistant layer is seven times higher than that of the copper plate, and the high temperature wear resistance is increased by

nearly two times. Ke et al. [16] successfully prepared Ni60 coating on the surface of copper plate by using plasma cladding. The coating and the copper plate present an excellent metallurgical bond. These studies simply characterize the microstructure and properties, and there is a lack of deep discussion and analysis on how to obtain high-quality coatings. Our group's previous research also successfully deposited Ni60A alloy on copper plate by plasma cladding (Figure S1 in Supplementary Materials). The effect of process parameters on the quality of Ni60A coating on copper surface has been investigated. However, the previous studies were all based on the surface strengthening of the copper plate. It is still a major challenge to develop a commercial fabrication process for coating the large-size curved copper components with high efficiency. The tuyere small sleeve is a large and complex curved copper component with a diameter of 400 mm, a length of 600 mm and a weight of about 210 kg (Figure 1b). Therefore, from a small-sized copper plate to actual large-size tuyere small sleeve, it is necessary to carry out a pilot cladding experiment on the surface of a transition pipe (Figure 1a).

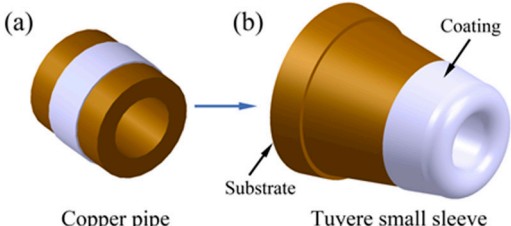

**Figure 1.** Development sequence of plasma cladding wear-resistant coating on copper substrates with different shapes.

In addition, due to the high thermal conductivity of copper, the heat provided by the plasma arc to the copper can be quickly dissipated, making it difficult to form a proper molten pool, especially for the plasma cladding of large-sized copper components. To overcome this problem, one commonly used method is a preheating substrate [17,18]. Among various heating methods, induction heating has the advantages of high-speed and local heating, which has potential application value in the preheating of copper surfaces [19]. How to achieve the dynamic matching control of plasma cladding and substrate preheating is the key and difficult point to obtain high-quality coating.

Developing a high-efficiency cladding system on a large-complex-curved surface, the key issue that needs to be solved first is to explore its cladding law. In this study, plasma cladding coupled with induction heating was exploited to fabricate Ni60A coating on copper pipe. The copper pipe has the same wall thickness as the tuyere small sleeve, which is used as a simulant to verify the feasibility of the system. To clarify the characteristics of cladding on the curved surface, the evolution of the microstructure and properties of the coating from the head to the tail around the copper pipe was systematically investigated. The regulation mechanism for obtaining high-quality coatings were discussed and summarized, which provides a theoretical and technical basis for the surface cladding of large-size tuyere small sleeve.

## 2. Experimental Procedures

### 2.1. Materials and Sample Preparation

Pure copper pipe (99.99% wt.% Cu) with an outer diameter of 50 mm, inner diameter of 30 mm and a length of 50 mm were used as the substrate, as shown in Figure 2a. Before cladding, the copper pipe was polished with 800 mesh sandpaper and cleaned with ethanol to remove oxide scale and oil. Ni60A powder was used as cladding material, which has a good compatibility with copper [20–22]. Table 1 shows the chemical composition of the Ni60A powder. The powder is spherical with a size of 45–100 μm (Figure S2 in Supplementary Materials). To maintain good fluidity, the powder was placed in a drying oven and dried at 120 °C for 2 h.

The experiment was carried out on a numerical control plasma cladding equipment (PTA-400A, Shanghai Benxi Electromechanical, Shanghai, China) with powder feeding system. The cladding device and the preparation process of the coating are described in detail in Section 3.1. The appearance of the Ni60A coating prepared on copper pipe by plasma cladding is shown in Figure 2b. To study the evolution of the microstructure and properties of the coating along the copper pipe, the whole coating was cut into 6 equal parts from the head to the tail. The samples were denoted as S1, S2, S3, S4, S5 and S6, as shown in Figure 2c.

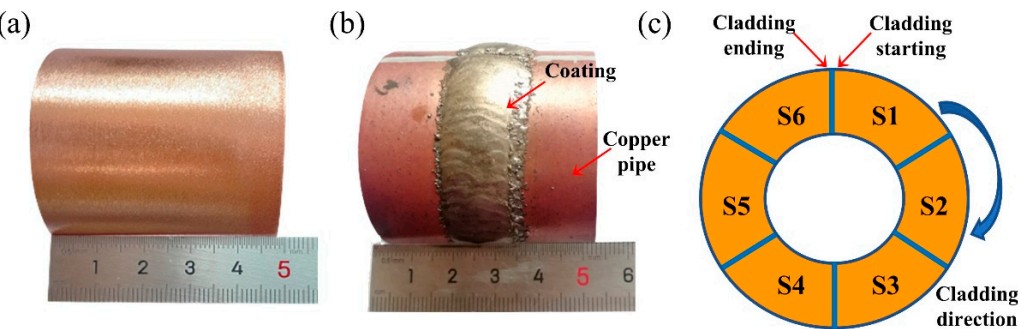

**Figure 2.** (**a**) Original morphology of copper pipe, (**b**) morphology of Ni60A coating on copper pipe by plasma cladding and (**c**) sampling method.

**Table 1.** Chemical composition (wt.%) of the Ni60A powder.

| Cr | C | Si | B | Fe | Ni |
|---|---|---|---|---|---|
| 15.0 | 0.8 | 4.0 | 3.5 | 8.0 | Balance |

## 2.2. Characterization

The samples were sectioned and then polished for microstructural examination. Microstructural analysis of the sample was performed using FEI Nova Nano SEM 450 scanning electron microscope (SEM) equipped with energy-dispersive spectrometer (EDS). The observation position of the microstructure is at the junction surface of two adjacent specimens. The geometric dimensions of the coating were measured by SEM. The phase composition of the sample was identified by X-ray diffraction (XRD, Bruker-AXS D8 Advance, Billerica, MA, USA) using Cu K$\alpha$ diffraction ($\lambda$ = 1.54056 Å) with a scanning angle ranging from 10° to 90° and a scanning speed of 6°/min. The area of the primary carbides was counted by Image-Pro Plus 6.0 (Media Cybernetics, Inc., Rockville, MD, USA) commercial software.

The hardness along the depth of the cross-section was measured using a microhardness tester (HVS-1000M) at a load of 100 g and a loading time of 10 s. A wear test was conducted on a pin-on-disc friction and wear tester (MMUD-5B) at room temperature (Figure S3 in Supplementary Materials). The pin was the test sample with a dimension of $\varphi$4 mm × 10 mm. The disc was the friction pair (GCr15 steel) with a dimension of $\varphi$45 mm × 4 mm and a hardness of 60 HRC. Before the test, the samples were polished with 1500 mesh sandpaper to maintain the same roughness. The wear conditions were a load of 50 N, a rotating speed of 100 r/min and a test time of 15 min. The wear resistance of Ni60A coating and copper substrate was evaluated by the mass loss, which was weighed by an electronic balance with an accuracy of 0.1 mg. The value of mass loss was taken as the average of three repeated test results. The friction coefficient was continuously recorded by an online data acquisition system from the wear tester. After the wear test, the worn morphology was observed by SEM to analyze the wear mechanism.

## 3. Results and Discussion

### 3.1. Fabrication of Ni60A Alloy on Copper Pipe

Figure 3a shows the schematic diagram of the cladding system. To reduce the heat loss of the plasma arc on the surface of the copper pipe, an induction preheating system was introduced, which can preheat the copper pipe to the required temperature in a few minutes. The induction preheating system included an induction power supply, an infrared thermometer (CTlaser MT, Optris, Germany) and a proportional integration differentiation (PID) controller. To ensure the accuracy of temperature measurement, the parameter of emissivity needed to be set to the emissivity of copper (the value is 0.8). The surface temperature of the cladding zone measured by the infrared thermometer was fed back to the PID controller, so as to achieve the dynamic adjustment of heating power. Most importantly, the infrared thermometer was fixed on the bracket of the plasma welding gun, which can ensure that the temperature measurement point and the cladding point were always synchronized. The temperature curve of the cladding zone during the cladding process was recorded by a computer.

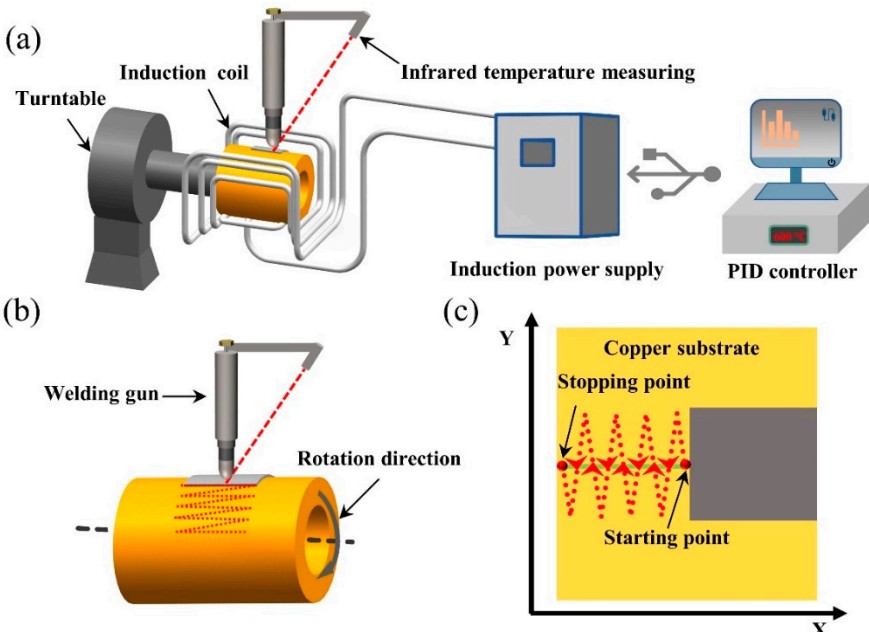

**Figure 3.** (**a**) Schematic diagram of the cladding system, (**b**) detail view of the swinging arc plasma cladding process and (**c**) magnified view of swinging arc path.

To improve the efficiency of the plasma cladding, a program of swing arc was used, as shown in Figure 3b. That is, the copper pipe kept a high-speed rotation, while the plasma welding gun swung back and forth perpendicular to the direction of rotation. The magnified view of the swing arc process is shown in Figure 3c, where $Y$ is the direction of the swing arc, and $X$ is the direction of the rotation direction. The actual cladding path can be seen as a "Z" shape. During the cladding process, the perfect overlap can only be achieved by matching the rotation speed of the copper pipe and the swing speed of the welding gun. The preliminarily optimized cladding process parameters are shown in Table 2. Among them, the preheating temperature was obtained based on the previous experimental results on the copper plate (Figure S1 in the Supplementary Materials). The basic principle of process parameter optimization is to ensure that the powder can be completely melted and fused with the copper substrate. Under these process parameters, the cladding efficiency of the coating (Figure 2b) can reach 32.72 mm$^2$/s, which is about 2.7 times higher than that of a single-track cladding of copper pipe (12 mm$^2$/s). The detailed calculation method of cladding efficiency is shown in the Section 4 of the Supplementary Material.

**Table 2.** Cladding parameters of swing arc cladding.

| Parameter | Value |
|---|---|
| Preheating temperature (°C) | 600 |
| Rotating speed of copper pipe (mm/s) | 2.18 |
| Current (A) | 140 |
| Powder feeding speed (g/min) | 19.74 |
| Nozzle distance (mm) | 10 |
| Plasma gas flow rate (L/min) | 3 |
| Shielding gas flow rate (L/min) | 9 |
| Powder feeding gas rate (L/min) | 3 |
| Swing speed of welding gun (mm/min) | 2500 |
| Swing arc width (mm) | 15 |

*3.2. Microstructure Analysis*

Figure 4 shows the variation trend of the temperature, height, bath depth, and dilution ratio of the coating from S1 to S6. The dilution ratio of the coating was calculated according to the formula in reference [23]. As shown in Figure 4, the surface temperature of the cladding zone of the copper pipe increases continuously with the progress of the cladding. The height of coatings shows a decreasing trend, while the bath depth and dilution ratio gradually increased from S1 to S6. This indicates that the coating appears inconsistent at the head and tail even if the process parameters are constant. The heat balance of the substrate surface can explain this inconsistency. Actually, the surface temperature of the cladding zone of copper pipe depends on the balance of heat input and output. The input heat mainly comes from the induction heating system and plasma arc, and the output heat is mainly the heat dissipation of the copper pipe. Theoretically, the surface temperature is constant under the condition of fixed process parameters. However, the plasma arc continuously heated the copper pipe, while the heat cannot be dissipated in time during the cladding process. Therefore, the original heat balance on the surface of the cladding zone was broken, resulting in a continuous increase in the surface temperature. The higher temperature corresponds to the larger proportion of melted copper, leading to an increase in the bath depth. In thermodynamics, the surface tension of the melt exhibits a negative temperature coefficient [23]. The higher temperature results in the smaller surface tension, the molten pool can be easier to spread to the edge, thus increasing the dilution ratio of coating.

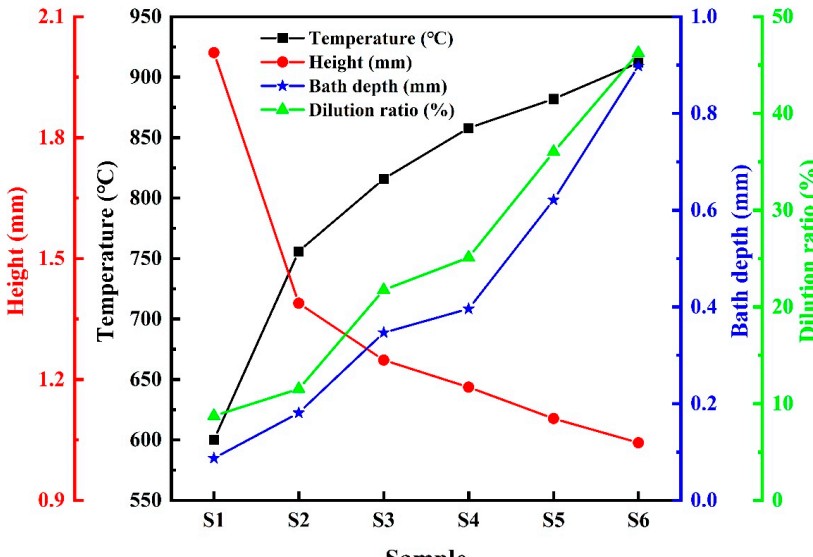

**Figure 4.** Variation curves of temperature, height, bath depth and dilution ratio with the progress of cladding.

Figure 5 shows the SEM images of microstructure of the coatings from S1 to S6. It can be seen that the coatings have no defects such as surface cracks or pores and the top, middle and bottom areas of the coating always show different microstructures. A large number of fine and uniform black granular phases and gray acicular phases are distributed on the top and bottom of the coating. Compared with the top and the bottom part, the phases in the middle are significantly larger in size. This phenomenon is related to the temperature gradient of the coating. During the cladding process, the top of the coating was cooled by the airflow and had a large temperature gradient. Meanwhile, the bottom also endured a large temperature gradient due to the high thermal conductivity of the copper substrate. Therefore, the solidification time of the top and bottom of the coating was shorter than that of the middle part. The precipitated phase in the middle of the coating had enough time to grow up, resulting in a larger size.

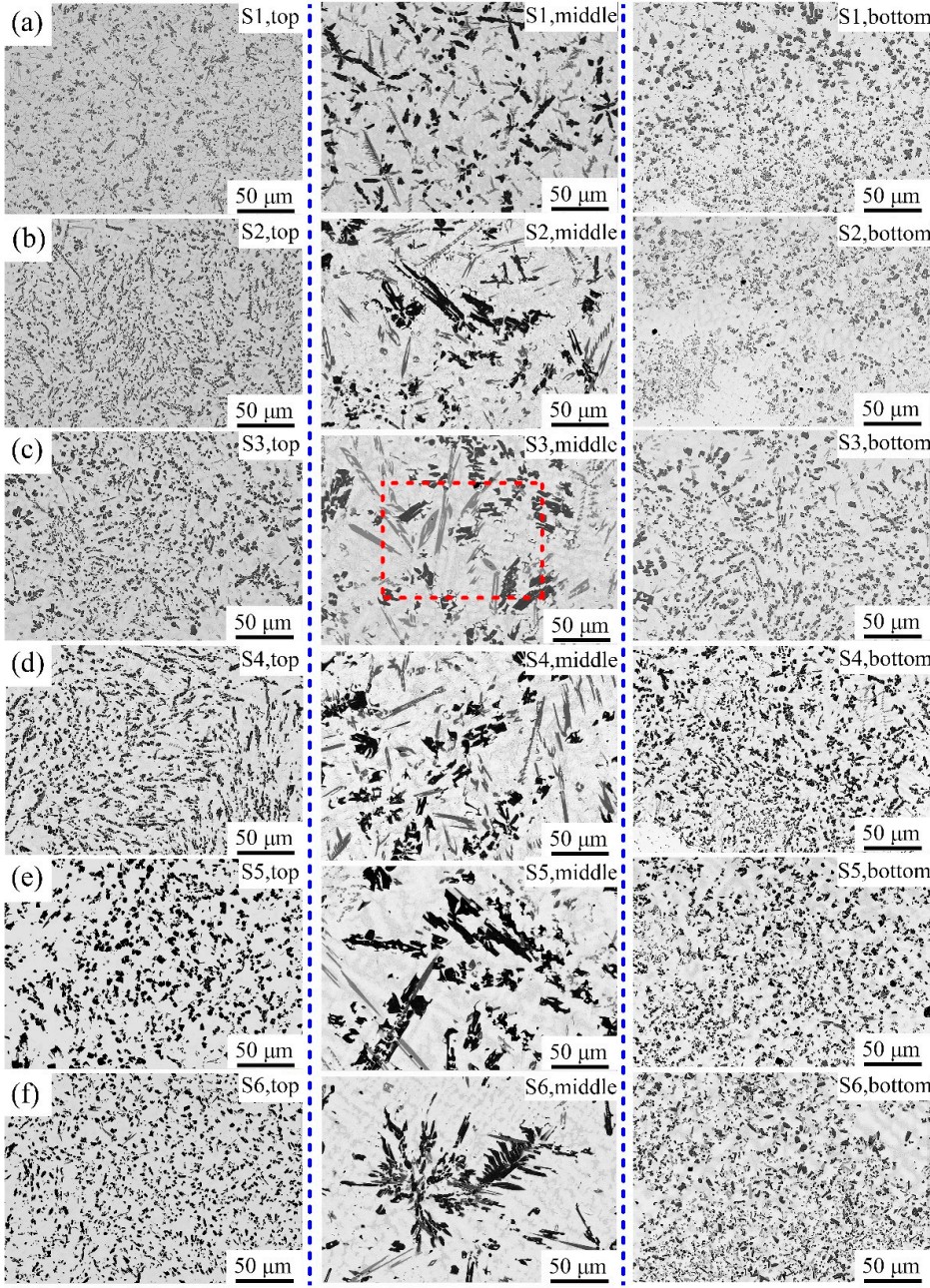

**Figure 5.** SEM images of the microstructure of the coatings from S1 to S6: (**a**) S1, (**b**) S2, (**c**) S3, (**d**) S4, (**e**) S5, and (**f**) S6.

The magnified SEM micrograph corresponding to the area marked by a small rectangle in Figure 5c is shown in Figure 6a. It can be found that the microstructure mainly includes four phases: black blocky phases (indicated by Spot 1), dark gray strip-shaped phases (indicated by Spot 2), light gray dendritic phases (indicated by Spot 3) and a light gray matrix (indicated by Spot 4). The EDS analysis results of these points are shown in Figure 6b–e, and the atomic percentage of each marked point is listed in Table 3.

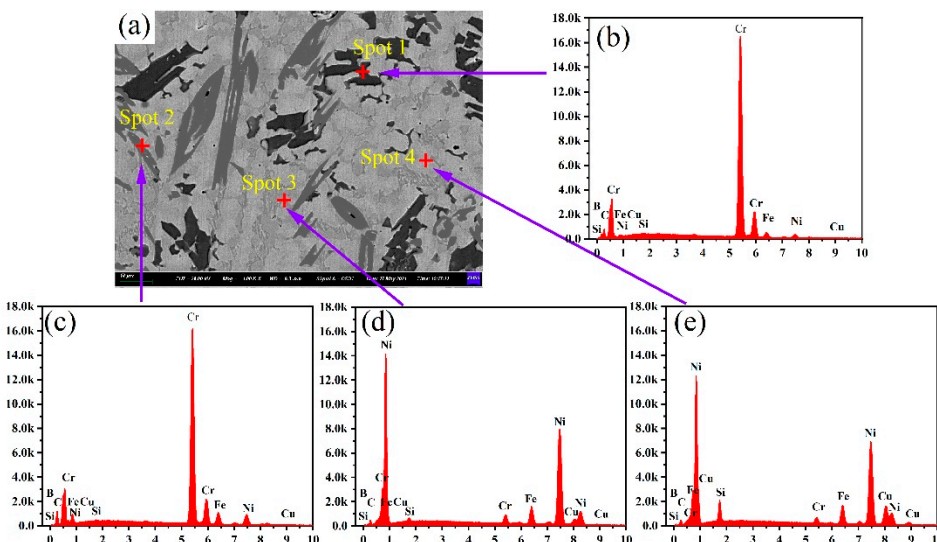

**Figure 6.** SEM micrograph and EDS analysis of Ni60A coating: (**a**) the magnified SEM micrograph of the area marked by a small rectangle in Figure 5c, (**b–e**) the EDS analysis results of spots 1–4.

**Table 3.** EDS spot analysis results of each point marked in Figure 6 (at.%).

| Spot | B | C | Si | Cr | Fe | Ni | Cu |
|---|---|---|---|---|---|---|---|
| 1 | 35.17 | 20.36 | 0.09 | 41.24 | 1.61 | 1.25 | 0.28 |
| 2 | 9.13 | 30.12 | 0.17 | 50.48 | 4.63 | 5.01 | 0.46 |
| 3 | 8.11 | 9.14 | 14.52 | 1.75 | 3.12 | 53.05 | 10.31 |
| 4 | 12.16 | 20.98 | 5.19 | 1.48 | 5.31 | 42.87 | 12.01 |

The XRD pattern of the Ni60A coating is shown in Figure 7a. According to the PDF comparison results of diffraction peaks, the phase constituents of Ni60A coating include $CrB$, $Cr_{23}C_6$, $Ni_3Si$ and $\gamma$-(Cu, Fe, Ni). Combined with the results of EDS analysis in Figure 6, the black blocky phase can be identified as $CrB$ and the dark gray phase as $Cr_{23}C_6$. They are the primary reinforcements of the coating. The light gray dendritic phase is $Ni_3Si$, and the light gray matrix is $\gamma$-(Cu, Fe, Ni). Due to the dissolution of Cu, the diffraction peak of $\gamma$-(Cu, Fe, Ni) shifts to a small angle and the diffraction peak intensity becomes higher from S1 to S6. Figure 7b shows the X-ray diffraction spectrum of the S2. There is no difference in the phase types of the top, middle and bottom of the coating. The diffraction peak of bottom is shifted to the left relative to that of the middle and top, which may be due to the higher copper content at the bottom of the coating.

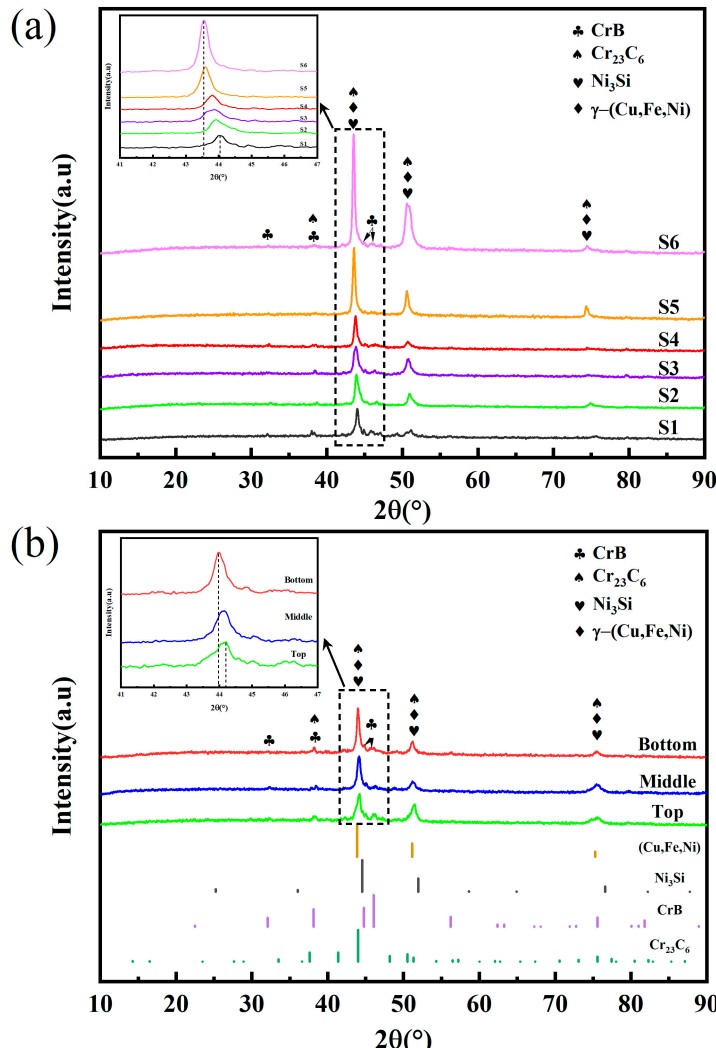

**Figure 7.** X-ray diffraction patterns of the Ni60A coating: (**a**) S1–S6, (**b**) different parts in the S2.

The principle diagram of the coating solidification process is shown in Figure 8. The melting point of CrB (2760 °C) is higher than that of $Cr_{23}C_6$ (1520 °C), so it firstly precipitates form the molten pool during the plasma cladding process. Due to the incompatibility of Cu and Cr, Cu atoms were repelled into the remaining liquid phase (Figure 8a). As the temperature of the molten pool further decreases, $Cr_{23}C_6$ becomes needle-like phase precipitation. Subsequently, the atom of Si and Ni formed $Ni_3Si$ and then the γ-(Cu, Fe, Ni) solid solution formed in the remaining liquid phase (Figure 8b). However, from Figure 5, it can also be seen that the CrB phases have a slight tendency to grow up and reunite with the progress of the cladding. Meanwhile, the area of the carbide phase has a significant change during the solid solution process. The statistical results in Figure 9 prove that the area fraction of $Cr_{23}C_6$ has a decreasing trend (The area fraction was studied from 10 randomly selected fields in the middle of each sample). This is mainly related to the longer liquid time under the high surface temperature of cladding zone. In the longer liquid time, the CrB more easily adsorbed a large amount of Cr atom, which not only results in the growth and agglomeration of CrB (Figure 8c) but also leads to a certain decrease in the area fraction of $Cr_{23}C_6$ (Figure 8d).

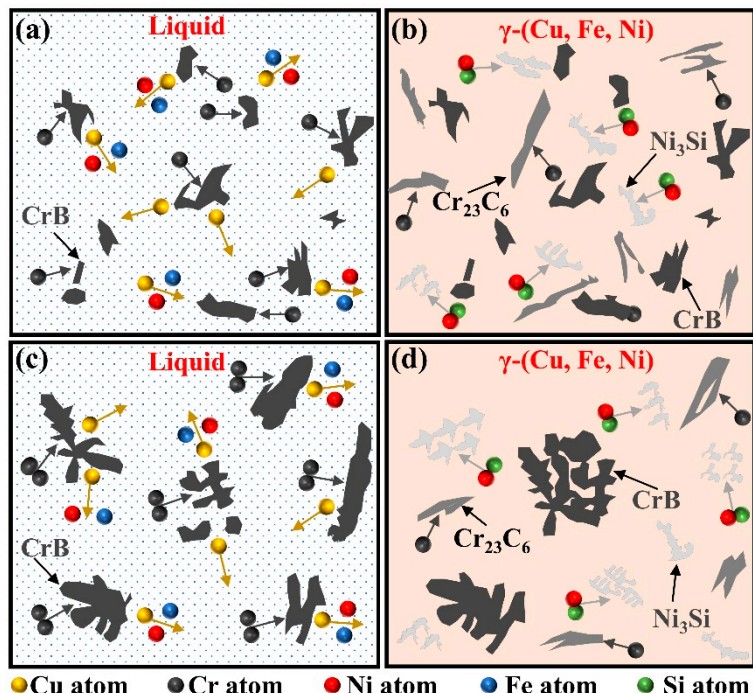

**Figure 8.** Principle diagram of the solidification process of the coating: (**a**) the nucleation and growth of CrB in shorter liquid time, (**b**) the formation of remaining phases in shorter liquid time, (**c**) the nucleation and growth of CrB in longer liquid time and (**d**) the formation of remaining phases in longer liquid time.

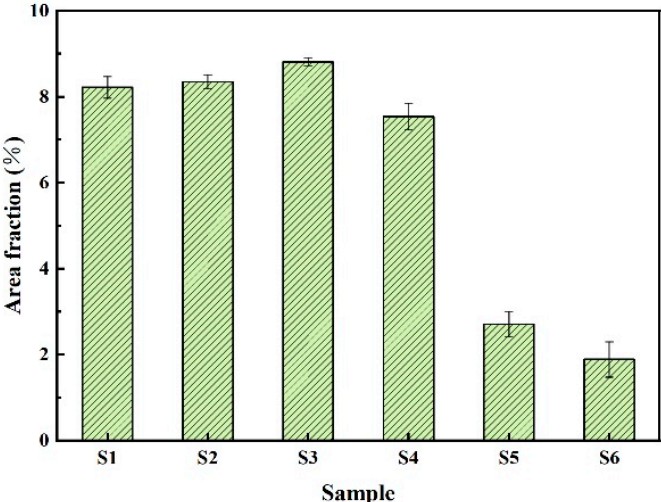

**Figure 9.** The area fraction of $Cr_{23}C_6$ phase.

Figure 10 shows the EDS line scan results along the copper substrate to the top of the coating (S1–S6). The distribution of several main elements (Cu, Ni and Cr) is specially displayed. It can be seen that there is an interface transition zone between the coating and the substrate. The Cu element presents a gradient decrease at the interface transition zone, while the content of Cu element inside the coating is basically the same. The content change of the Ni element is opposite that of the Cu element. The content of Cr element fluctuates in the transition zone, which is related to the dispersion distribution of CrB and $Cr_{23}C_6$ precipitates. Therefore, the coating prepared by plasma cladding can be regarded as an in situ synthesized gradient coating. With the increase of cladding temperature, the copper content in the coating and the range of the interface transition zone has an

increasing trend. Especially for S5 and S6, the copper content and its distribution have the most significant changes. Because of the good compatibility of copper and Ni-based alloys, the molten copper and Ni60A alloy can be fully mixed. At higher cladding temperatures, the molten pool has a lower viscosity, which leads to a stronger stirring effect of the molten pool, thus resulting in more copper distribution in the coating [24]. In addition, due to the dilution effect of copper, the composition of the entire coating is changed. Therefore, the coating prepared by the cladding method cannot be simply regarded as a pure alloy, but a composite material of Ni60A and copper. For a high-quality coating, there should be no large sudden change in composition. Therefore, further optimization of process parameters is required to control the coating at least between S1–S4.

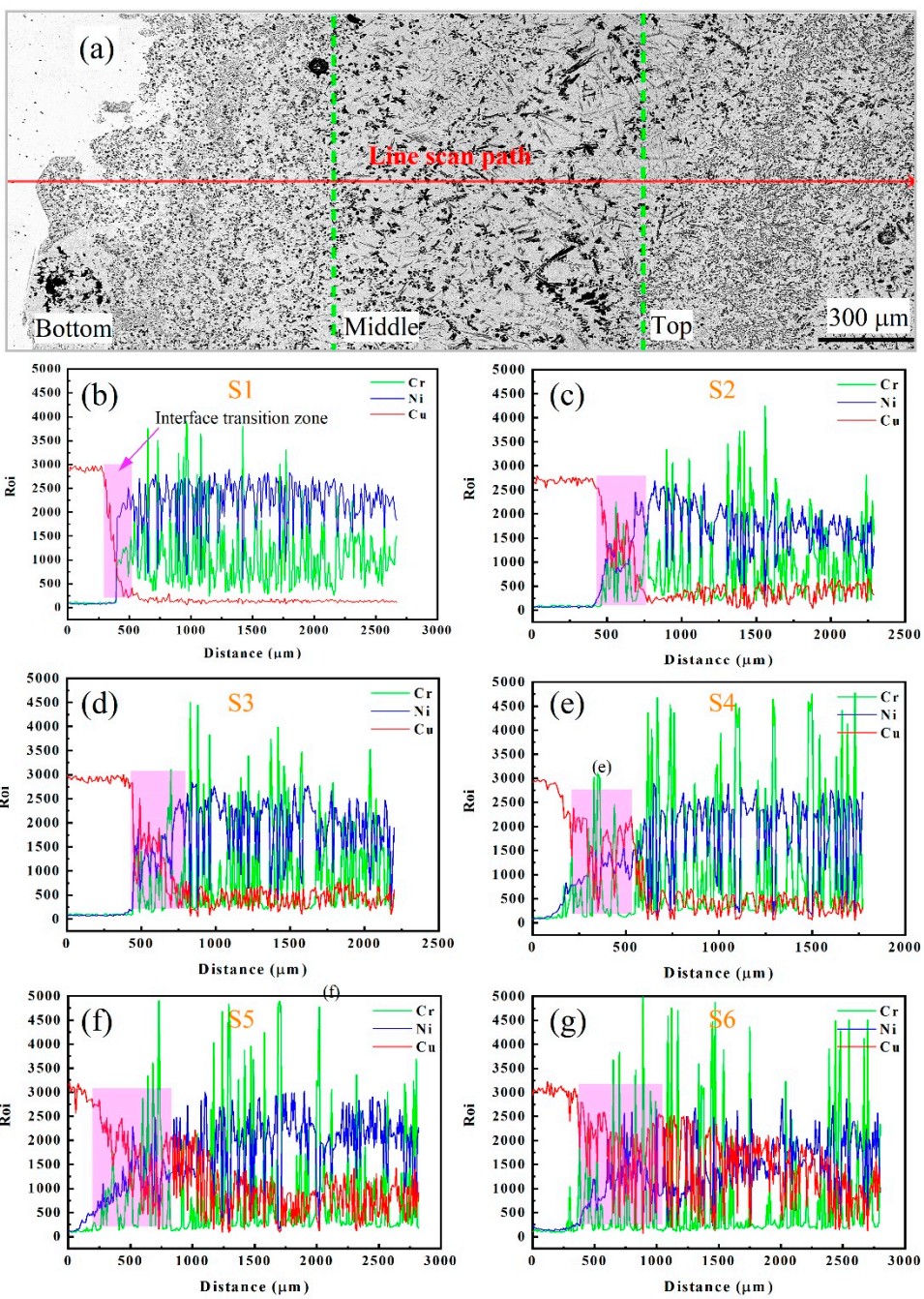

**Figure 10.** EDS line scan results of S1–S6 along the direction from the copper substrate to the coating: (**a**) SEM micrograph from copper substrate to the top of the coating, (**b**–**g**) the EDS line scan results along the copper substrate to the top of the coating (S1–S6).

### 3.3. Microhardness

Figure 11 gives the microhardness variation tendency of the coatings (S1–S6) along the direction of cross-section depth. The hardness value inside the coating (about 840 HV–450 HV) is significantly higher than that of the copper substrate (about 65 HV), which mainly attributed to the presence of precipitated phases such as CrB, $Cr_{23}C_6$, $Ni_3Si$ and $\gamma$-(Cu, Fe, Ni) phase in the coating (Figure 6). They increase the hardness of the coating under the influence of dispersion strengthening of CrB and $Cr_{23}C_6$ phases and the solid-solution strengthening of $\gamma$-(Cu, Fe, Ni) phase. Moreover, $Ni_3Si$ also has the effect by hindering the dislocation motion [25]. The hardness value near the copper substrate shows a gradient decline trend, which is basically consistent with the change trend of copper content. From S1 to S6, the hardness of the coating gradually decreased. Compared with the S1, the average hardness of the S6 coating is reduced by about 1.5 times, attributed to the increase of the dilution ratio. In addition, due to the low dilution ratio of S1, the hardness transition zone is not obvious. From S2 to S6, the range of the hardness transition zone gradually increases, which corresponds to the gradient change distance of the Cu element in each sample (Figure 10).

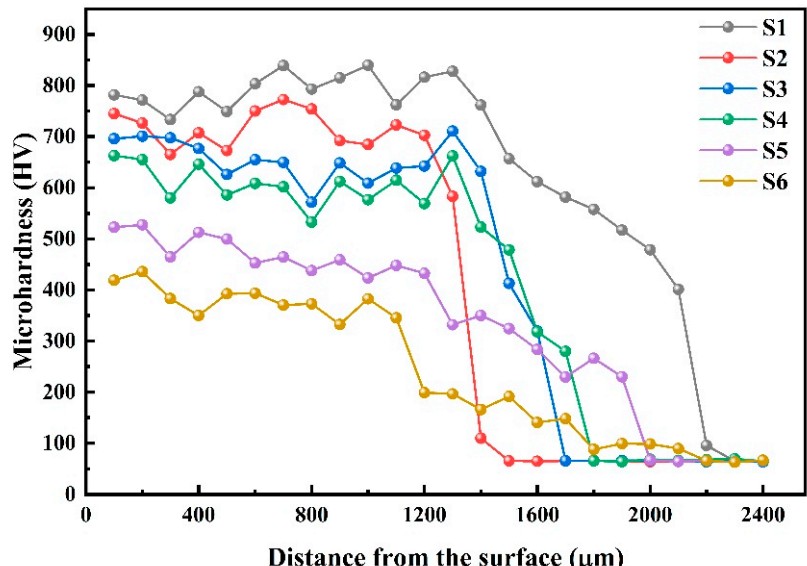

**Figure 11.** Microhardness distribution of different samples along the depth of the coatings.

### 3.4. Wear Resistance

Figure 12a shows the friction coefficient of the pure copper and Ni60A coatings (S1–S6). Obviously, the average friction coefficient of pure copper (0.264) is higher than that of the coatings. Among all samples, S5 and S6 show higher friction coefficients, while the friction coefficients of other samples are basically the same. Generally, a material with a low coefficient of friction indicates better wear resistance. The wear resistance of the coating is better than that of pure copper mainly due to the good bearing capacity of carbide- and boride-reinforced particles in the coating. These hard particles can reduce the contact area between the friction surface and its surface, thus reducing the smearing effect [26].

Figure 12b shows the mass loss of the pure copper and Ni60A coatings (S1–S6). It can be seen intuitively that the mass loss of the coatings is greatly reduced compared to the copper substrate. Among all samples, the wear resistance of S1 is about 4.5 times that of the copper substrate. As the cladding progresses, the mass loss is slightly reduced from S1 to S4, while of the S5 and S6 has a significant in the mass loss. Generally, under the same surface roughness, the small amount of wear of the coating is mainly due to its higher hardness. The mass loss of material is directly proportional to the coefficient friction and inversely proportional to the hardness of the material [27]. From the head to the tail

of coating, the higher the dilution ratio, the closer the performance of the coating is to pure copper.

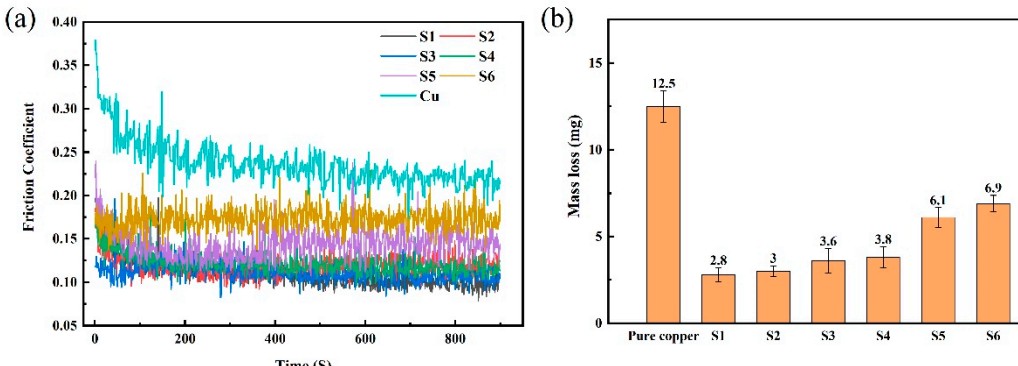

**Figure 12.** Wear test results of pure copper and Ni60A coatings (S1–S6): (**a**) friction coefficient and (**b**) mass loss.

Figure 13 shows the SEM morphologies from worn surfaces of Ni60 cladding (S1–S6). It can be seen from Figure 13a–c that there is no spalling or adhesion phenomenon on the surfaces of the S1–S3 coatings, only some grooves formed by plowing, which is a typical abrasive wear. As shown in Figure 13d, some spalling pits appear on the surface of the S4 coating, and grooves can still be found on its surface. This indicates that the wear mechanism of S4 coating is abrasive wear and adhesive wear. However, the S5 and S6 coatings suffer a severe adhesive wear due to the large number of spalling pits on the surface. Combined with Figure 11, the increase in mass loss of coating mainly depends on the severity of adhesive wear. From the analysis of the hard phase of the coating, the change of the wear mechanism is related to the decrease of $Cr_{23}C_6$.

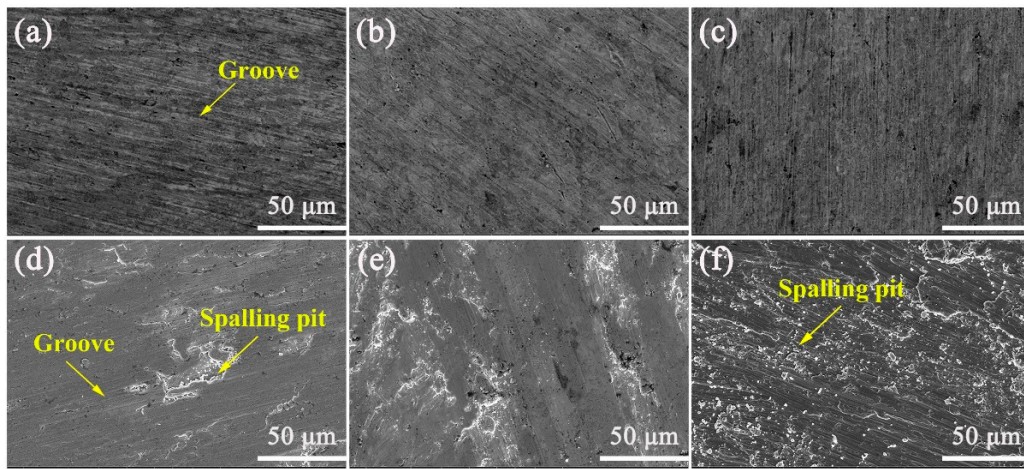

**Figure 13.** SEM micrographs of the worn surface: (**a**) S1, (**b**) S2, (**c**) S3, (**d**) S4, (**e**) S5 and (**f**) S6.

### 3.5. Regulation Mechanism of Plasma Cladding Process

As shown in Figure 14, the fundamental reason for the evolution of dilution ratio and properties is the change in the surface temperature of the cladding zone on copper pipe. A surface temperature of 850 °C can be regarded as the critical temperature for a sudden change of properties. Before and after this temperature, the wear mechanism of the coating completes the transition from abrasive wear to abrasive wear and adhesive wear to adhesive wear, thus significantly reducing the wear resistance. Therefore, controlling the surface temperature of the cladding zone below 850 °C can basically maintain the good quality of the coating.

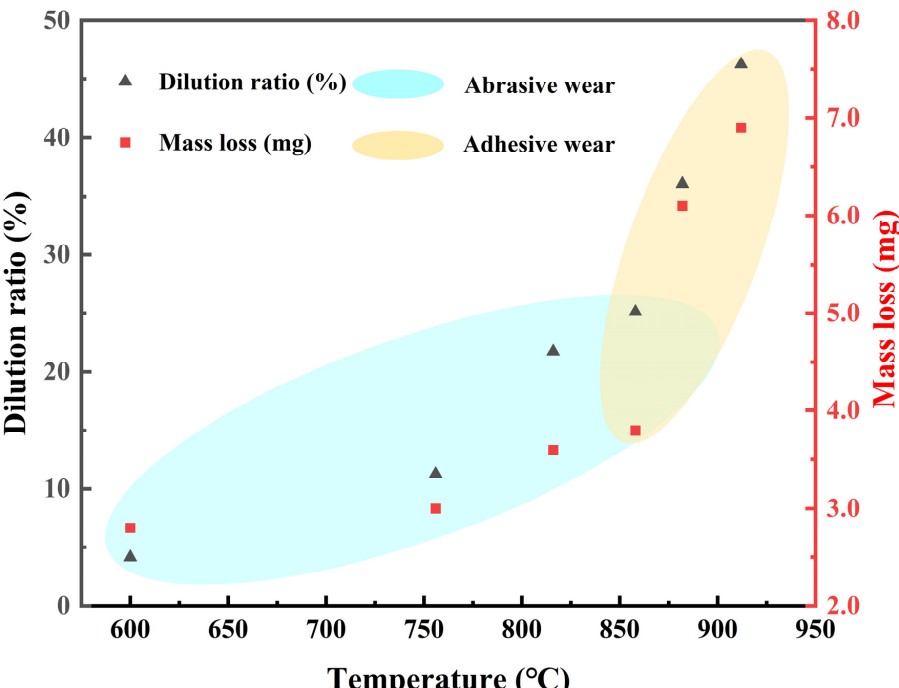

**Figure 14.** Summary diagram of the relationship between cladding temperature, dilution ratio and mass loss.

In addition, as discussed in Section 3.1, the temperature of the cladding zone (*T*) is directly related to the heat balance on the surface of the copper pipe, which can be expressed as:

$$T \sim Q_1 + Q_2 - Q_3 = f(P) + f(I, v) - f(V, \lambda) \tag{1}$$

where the $Q_1$ is the heat provided by induction heating, which is related to the heating power (*P*), denoted as $f(P)$. $Q_2$ is the heat provided by the plasma arc, which is related to the cladding current (*I*) and the rotation speed of the copper pipe (*v*, corresponding to the residence time of the plasma arc on the copper pipe), denoted as $f(I, v)$. $Q_3$ is the heat dissipation of the copper pipe, which is related to the volume (*V*) and the thermal conductivity ($\lambda$) of the copper pipe, denoted as $f(V, \lambda)$. During the cladding process, the plasma arc continuously heats the copper pipe, causing the heat input to be greater than the heat output, resulting in a continuous increase in the temperature of the cladding zone (T). To ensure a basically constant temperature to make the coating as uniform as possible, the heat input of the copper pipe must be gradually reduced during the cladding process, thus achieving a balance with heat dissipation. As a process before cladding, the heat input of the preheating system ($Q_1$) cannot be adjusted during the cladding process. Therefore, the only way to reduce the total heat input is to reduce the heat input of the plasma arc ($Q_2$). However, adjusting the rotation speed of the copper pipe (*v*) during the cladding process is inconvenient, because the rotation speed needs to be matched with the swing speed of the welding gun. As a result, gradually reducing the cladding current (*I*) during the cladding process become the only way to make the coating as uniform as possible and further improve the quality of the coating, which can be easily achieved in the current numerical control plasma cladding system. As shown in Figure 15, by adjusting the cladding current in stages to control the surface temperature of the cladding zone in the range of 600–850 °C, the dilution rate and mass loss of the cladding layer have little fluctuation as a whole.

For the actual tuyere small sleeve with large-complex-curved shape, according to the above discussion, reducing the current in stages to keep the surface temperature of the tuyere small sleeve in the range of 600–850 °C is the key to obtain a high-quality coating. Furthermore, based on the cladding on copper pipe, it is first necessary to increase the size

of the induction heating coil (Figure 16) and the power of the induction power supply to ensure that the small sleeve can be heated to the required preheating temperature (600 °C).

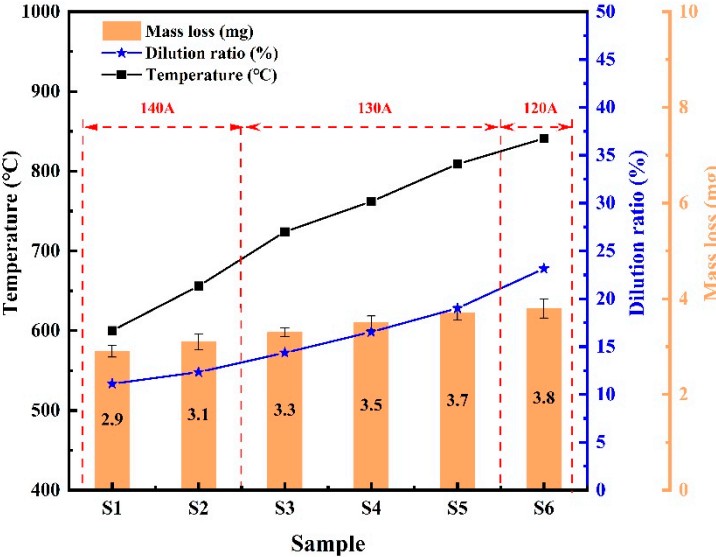

**Figure 15.** Diagram of the relationship between cladding temperature, dilution ratio and mass loss after adjusting the cladding current in stages.

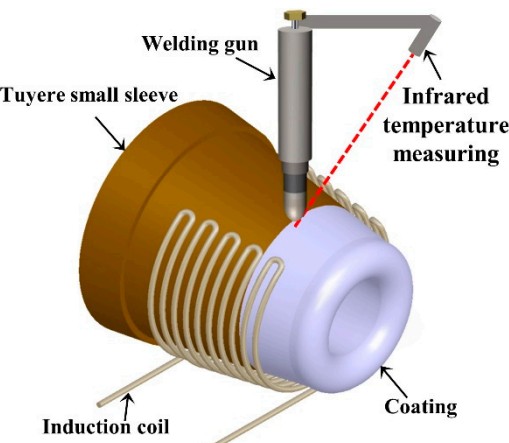

**Figure 16.** Schematic diagram of cladding process for tuyere small sleeve.

## 4. Conclusions

(1) A plasma cladding coupled induction heating process was developed to fabricate Ni60A coating on curved surface of pure copper pipe. The cladding system has the advantages of high efficiency and real-time temperature control. By setting the swing arc program, the cladding efficiency was as high as 32.72 mm$^2$/s, which is 2.7 times higher than that of a single-track cladding of copper pipe (12 mm$^2$/s).

(2) The surface temperature of the cladding zone on copper pipe continues to increase, which leads to an increase in the dilution ratio of the coating from the head to the tail. This is because the heat dissipation of the copper pipe is less than the heat input during the cladding process. The size of CrB phase increases to a certain extent and tends to agglomerate. The precipitation phase of $Cr_{23}C_6$ grows from the fine needle phase to the long strip phase, and its content shows a decreasing trend. The coating can be regarded as an in situ synthetic gradient coating due to the gradient change of composition.

(3) The hardness and wear resistance of the coating shows a significant decrease when the copper surface temperature reaches above 850 °C. Before and after this temperature, the wear mechanism of the coating presents a transition law of abrasive wear → abrasive wear and adhesive wear → adhesive wear. From the head to the tail of the coating, the wear resistance changed from 4.5 times to 1.8 times that of pure copper.

(4) It can be concluded that the plasma cladding coupled induction heating is an effective means for strengthening the surface of curved copper components. The key is to reduce the cladding current in stages to keep the temperature of the cladding zone on the surface of the copper pipe within the range of 600–850 °C. Based on the preliminarily optimized cladding parameters of copper pipe, it is necessary to increase the size of the induction heating coil and the power of the induction power supply to ensure that the small sleeve can be heated to the required preheating temperature (600 °C).

**Supplementary Materials:** The following are available online at https://www.mdpi.com/article/10.3390/coatings11091080/s1, Figure S1: Appearance of Ni60A coatings on copper plate under different preheating temperature: (**a**) room temperature, (**b**) 300 °C, (**c**) 400 °C, (**d**) 500 °C, (**e**) 600 °C, (**f**) 700 °C and (**g**) 800 °C., Figure S2: The SEM morphology of the Ni60A alloy powder, Figure S3: Schematic diagram of the wear test process, Figure S4: Appearance of single-track Ni60A coating on copper pipe, Figure S5: SEM micrograph of Ni60A coating, Figure S6: Cross-section SEM micrograph and the elemental mappings of cross-section of the Ni60A coating (S3), Table S1: Parameters of single-track plasma cladding process.

**Author Contributions:** Formal analysis, C.Z.; Investigation, D.B. and G.L.; Resources, X.L.; Software, Z.C.; writing—original draft, J.L.; writing—review & editing, Y.Z. All authors have read and agreed to the published version of the manuscript.

**Funding:** This work was supported by the National Natural Science Foundation of China and the Joint Research Fund of China Bao-Wu Iron and Steel Group Company Limited (Grant No. U1860108, U1860203), State Key Laboratory of Advanced Special Steel, Shanghai Key Laboratory of Advanced Ferrometallurgy and the Science and Technology Commission of Shanghai Municipality (Grant No. 19DZ2270200), the National Natural Science Foundation of China (Grant No. 51576164).

**Institutional Review Board Statement:** Not applicable.

**Informed Consent Statement:** Not applicable.

**Data Availability Statement:** Data is contained within the article.

**Conflicts of Interest:** The authors declare no conflict of interest.

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
