# Peer review of "Fabrication and Characterization of Ni60A Alloy Coating on Copper Pipe by Plasma Cladding with Induction Heating"

_coatings, doi:10.3390/coatings11091080_

Round 1

Reviewer 1 Report

The peer-reviewed article is devoted to improving the operational properties of the tuyere small sleeve. The article is very well structured. Contains a large amount of graphic material, which greatly helps the reader to understand the features of the research. The research used modern equipment. The research results are presented competently and in detail. The conclusions of the research are presented in detail and reflect the essence. I believe that this article can be published as presented with minor additions and corrections. 

1. Line 276 indicates that wear resistance is related to the hardness of the material. You can supplement this article with the results of measuring the hardness of the samples obtained.
2. In section "2.2. Characterization" the method for determining the coefficient of friction is not described in detail.
3. Lines 203-207 describe the phases. You can additionally show a photograph of the microstructure showing these phases. This will allow the reader to better understand the features of the microstructure.

Author Response

Point 1: Line 276 indicates that wear resistance is related to the hardness of the material. You can supplement this article with the results of measuring the hardness of the samples obtained.

Response 1: Thank you for your suggestion. The hardness results of the samples have been added (Fig. 11 in the revised manuscript, page 12).

Point 2: In section "2.2. Characterization" the method for determining the coefficient of friction is not described in detail.

Response 2: Thank you for your recommendation. The friction coefficient was continuously recorded by online data acquisition system of the wear tester. The description of the method of characterizing the friction coefficient has been supplemented (Lines 122-123 in the revised manuscript).

Point 3: Lines 203-207 describe the phases. You can additionally show a photograph of the microstructure showing these phases. This will allow the reader to better understand the features of the microstructure.

Response 3: Thank you for your comment. Fig. 6(a) in the original manuscript has shown the microstructure of various phases. To make these phases more clearly observed, the SEM image in Fig. 6(a) has been further magnified, as shown in Fig. S5 in the supplementary material.

Reviewer 2 Report

This manuscript can be accepted for publication in Coatings after major  revision:

  1. What is the thickness of the Ni60 A coating and how the coating thickness would affect the wear results?.
  2. Is it possible to observe the cross-section of the Ni60 A coating? If so, the EDS mappings on the cross-sectional images would be added to show the distribution of the elements throughout the coating thickness.
  3. From the XRD peaks shown in Fig.7, the number of peaks was not enough to identify four different phases in the coatings. At least, XPS spectra would be needed to approve the presence of these phases in the coatings. Also, It is strongly recommended that the authors include Powder Diffraction Files (PDF) they used for XRD peak identification
  4. A comparison of the wear results with the results available in the literature (if any) would enhance the novelty of this work.

Author Response

Point 1: What is the thickness of the Ni60 A coating and how the coating thickness would affect the wear results?

Response 1:

(1) The thickness of the Ni60A coating is the sum of height and bath depth (Fig. 4).

(2) In this study (discussion in section "3.4 wear resistance"), the main factors that affecting the wear resistance of the coating are the microstructure and chemical composition of the coating, rather than the thickness.

Point 2: Is it possible to observe the cross-section of the Ni60 A coating? If so, the EDS mappings on the cross-sectional images would be added to show the distribution of the elements throughout the coating thickness.

Response 2: Thank you. We have supplemented the EDS mappings of one coating sample along the thickness direction (Fig. S6 in the "Supplementary Materials"). The element distribution can be clearly seen.

Point 3: From the XRD peaks shown in Fig.7, the number of peaks was not enough to identify four different phases in the coatings. At least, XPS spectra would be needed to approve the presence of these phases in the coatings. Also, it is strongly recommended that the authors include Powder Diffraction Files (PDF) they used for XRD peak identification.

Response 3: Thank you for your suggestion. Commonly, the multi-peak overlap phenomenon is existed in the XRD pattern of Ni-based coating [1-4]. To prove the reliability of the data, the XRD pattern of the standard PDF card corresponding to these phases is supplemented in Fig. 7 (page 9 in the revised manuscript). The four phases can be well identified.

[1] K. Zhang, S.J. Wang, W.J. Liu, R.S. Long, Effects of substrate preheating on the thin-wall part built by laser metal deposition shaping, Appl. Surf. Sci. 317 (2014) 839-855.

http://dx.doi.org/10.1016/j.apsusc.2014.08.113.

[2] F. Liu, C.S. Liu, X.Q. Tao, S.Y. Chen, Laser cladding of Ni-based alloy on copper substrate, J. Univ. Sci. Technol. B. 13 (4) (2006) 329-332.

https://doi.org/10.1016/S1005-8850(06)60068-6.

[3] M.Y. Li, M.J. Chao, E.J. Liang, J.M. Yu, J.J. Zhang, D.C. Li, Improving wear resistance of pure copper by laser surface modification, Appl. Surf. Sci. 258 (4) (2011) 1599-1604.

https://doi.org/10.1016/j.apsusc.2011.10.006.

[4] Y.Z. Zhang, Y. Tu, M.Z. Xi, L.K. Shi, Characterization on laser clad nickel-based alloy coating on pure copper, Surf. Coat. Technol. 202 (24) (2008) 5924-5928.

http://dx.doi.org/10.1016/j.surfcoat.2008.06.163.

Point 4: A comparison of the wear results with the results available in the literature (if any) would enhance the novelty of this work.

Response 4: Thank you for your comment. The principle and conditions of the wear test in this study are different from those in other references, so they are not very comparable. Therefore, the evolution of wear resistance of the coating was discussed based on the comparison with that of the copper substrate under the same test conditions.

Reviewer 3 Report

Abstract -This section is well written and provides sufficient background for the reader and adequately summaries the main findings of the study.

Introduction-This section could be improved by reviewing more current study in the area to update the background of the study. Within the references cited 14 articles are more than five years old. Also a critical analysis of the literature and the state of art is required,

Methodology. The methods are clearly described and would allow for reproducibility of the the experiments if needed. Some of the content described in section 3.1 should actually have been covered in the methods section however it is suspected that it was only presented in section 3.1 to avoid repetition. 

Results -The result throughout the article appears consistent. The main thesis suggests that the thermal gradient across the samples modifies the microstructure and the mechanical performance of the clad layer. The microstructural images adequately represents that variation in the microstructure based on annular position. Similarly the wear assessment correlates those findings and indicates that samples at higher temperature have lower wear resistance. These findings are consistent with acceptable convention. 

Conclusions. The conclusions are consistent with t he stated objectives of the paper. 

Overall -the article is well written. Further improvements to the experimental design could shed light on the impact of processing parameters. Also analysis of the as receive powder would be important. 

Author Response

Point 1: Abstract -This section is well written and provides sufficient background for the reader and adequately summaries the main findings of the study.

Response 1: Thank you for your approval.

Point 2: Introduction-This section could be improved by reviewing more current study in the area to update the background of the study. Within the references cited 14 articles are more than five years old. Also, a critical analysis of the literature and the state of art is required.

Response 2: Thank you for your suggestion. Some older references have been replaced with new ones and some critical literature analysis has been supplemented on page 2 (Line 47-48) in the revised manuscript.

Point 3: Methodology. The methods are clearly described and would allow for reproducibility of the the experiments if needed. Some of the content described in section 3.1 should actually have been covered in the methods section however it is suspected that it was only presented in section 3.1 to avoid repetition.

Response 3: Thank you for your comments. Through a lot of preliminary exploration, the repeatability of the experimental results can be guaranteed.

One important highlight of this article is the development of a plasma cladding coupled with induction heating system. Therefore, the cladding system and fabricating process were described in detail in section 3.1, and it is only a brief mention of the equipment model in section 2.1.

Point 4: Results -The result throughout the article appears consistent. The main thesis suggests that the thermal gradient across the samples modifies the microstructure and the mechanical performance of the clad layer. The microstructural images adequately represent that variation in the microstructure based on annular position. Similarly, the wear assessment correlates those findings and indicates that samples at higher temperature have lower wear resistance. These findings are consistent with acceptable convention.

Response 4: Thank you for your approval.

Point 5: Conclusions. The conclusions are consistent with the stated objectives of the paper.

Response 5: Thank you for your comments.

Point 6: Overall -the article is well written. Further improvements to the experimental design could shed light on the impact of processing parameters. Also, analysis of the as receive powder would be important.

Response 6: Thank you for your approval.

Round 2

Reviewer 2 Report

The authors have revised the manuscript according to the reviewers comments. It can be accepted now